# On the role of initial velocities in pair dispersion in a microfluidic chaotic flow

Eldad Afik [1,2] & Victor Steinberg [1]

Chaotic flows drive mixing and efficient transport in fluids, as well as the associated beautiful complex patterns familiar to us from our every day life experience. Generating such flows at small scales where viscosity takes over is highly challenging from both the theoretical and engineering perspectives. This can be overcome by introducing a minuscule amount of long flexible polymers, resulting in a chaotic flow dubbed 'elastic turbulence'. At the basis of the theoretical frameworks for its study lie the assumptions of a spatially smooth and random-in-time velocity field. Previous measurements of elastic turbulence have been limited to two-dimensions. Using a novel three-dimensional particle tracking method, we conduct a microfluidic experiment, allowing us to explore elastic turbulence from the perspective of particles moving with the flow. Our findings show that the smoothness assumption breaks already at scales smaller than a tenth of the system size. Moreover, we provide conclusive experimental evidence that 'ballistic' separation prevails in the dynamics of pairs of tracers over long times and distances, exhibiting a memory of the initial separation velocities. The ballistic dispersion is universal, yet it has been overlooked so far in the context of small scales chaotic flows.

[1] Department of Physics of Complex Systems, Weizmann Institute of Science, Rehovot 76100, Israel. [2] Present address: Division of Biology and Biological Engineering, California Institute of Technology, Pasadena, CA 91125, USA. Correspondence and requests for materials should be addressed to E.A. (email: Eldad.Afik@gmail.com)

To truly appreciate how come many find elastic turbulence astonishing, we first have to realise that our intuition is based on scenarios where the flow is dominated by inertia, quantified by high values of the Reynolds number. When we stir sugar in a cup of coffee, we typically drive the liquid in circles using the tea-spoon, yet the flow quickly evolves into a three-dimensional (3D) chaotic one, tremendously accelerating the homogeneous distribution of the sweetener throughout the beverage. This mixing flow is a manifestation of the nonlinearity due to the inertia of the fluid taking over the viscous dissipation; the ratio of the two is estimated by the Reynolds number.

Now imagine a fly walking in honey or a bacterium swimming in water—one cannot expect any dramatic effects on the flow beyond a few 'bug' distance units away from it. When the typical velocities and length scales are small, corresponding to very low values of the Reynolds number, the flows of non-complex liquids —also known as 'Newtonian'—are dominated by dissipation. As a result they can be generically characterised as smooth and pre-dictable. So long as the driving force and the boundary conditions are steady, so will be the flow. A special class of geometries can induce 3D flows, which despite being steady in time, may lead to mixing[1–3]; these 'chaotic mixers' rely on patterned boundaries[2] or the vessel geometry[3] to continuously generate recurring diverging streamlines, which due to the low Reynolds remain fixed in space and time. Therefore, mixing in microfluidic devices is normally limited to diffusion.

Nevertheless, when even a minute amount of long flexible polymers, such as DNA and protein filaments, are introduced, the flow may develop a series of elastic instabilities which render it irregular and twisted. This flow—'elastic turbulence'[4–6]—which is chaotic in time, has been shown to drive efficient mixing in microfluidic devices as it can take place at extremely low values of the Reynolds number[7]; in the case of our experiment, more than

six orders of magnitude smaller than the critical value for inertial turbulence in a pipe[8]. It is exactly for this reason that even a fluid dynamics expert may be amazed when presented with the visual contrast between the mixing due to elastic turbulence and the expected separation between fluid layers in a laminar flow when the polymers are absent, as presented in (ref. [5], Fig. [1]) and (Figs 21–22 of ref. [9]); more background on elastic turbulence can be found in the review paper ref. [10] and the references therein.

Understanding transport phenomena at small scales is of importance and wide interest mainly for two reasons: first, much of the dynamics relevant for biology and chemistry takes place at these scales[5,11–13]; second, microfluidic devices are playing an important role in research and industrial technologies[2,14–17], often including complex fluids and flows whose dynamics still lack a universal description.

To achieve a fundamental understanding of mixing and transport phenomena, these need to be related and derived from their underlying microscopic level of description, at its simplest, the dispersion of pairs of particles[11,18,19]. Inspired by seminal works on turbulence beneath the dissipative scale, theoretical attempts to understand elastic turbulence rely on the assumptions that the velocity field is smooth in space[10,20,21], associating it with the class known as the 'Batchelor regime'[18,22]. For the dynamics of passive point-like tracers this means that the relative velocity between pairs is proportional to the distance separating them, with the upshot of exponential separation on average, asymptotically in time[11,18]; in Supplementary Note [1] we sketch how the asymptotic exponential pair separation prediction comes about.

The experimental study of pair separation dynamics in elastic turbulence, taking place inside a tiny tube, has been limited thus far as it poses technical challenges: first, the positions of tracers are needed to be resolved over long times and distances, in particular when the tracers get nearby to each other, whereas the

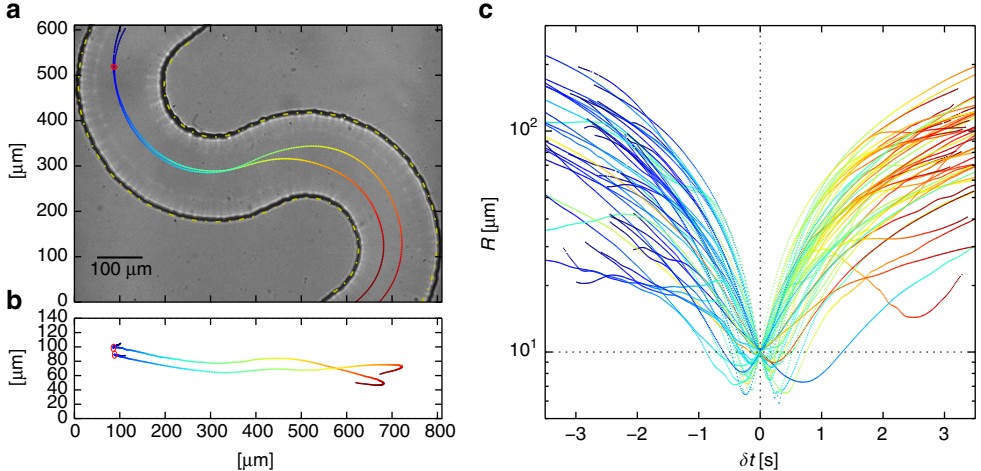

**Fig. 1** Pair dynamics example. The trajectories of two tracers are plotted in the *left panels*. The *right panel* shows a sub-sample of pair separation distances in the course of time. The figure outlines the analysis forming the ensembles of pairs, as well as demonstrates the chaotic nature of the flow as manifested by pairs; to develop the intuition and contrast with laminar flow, the reader is referred to, e.g., (Figs 21–22 of ref. [9]); several features of the mean flow in our case are manifested in the Eulerian representation in Supplementary Fig. [2], particularly the striking differences from Poiseuille-like laminar flows. **a** A projection on to the plane of the camera, which is imaging the channel from the bottom side (gravity pointing out of the panel towards the reader), overlaid on a bright field image of the observation window (further technical details are provided in Supplementary Fig. [1] and in the Methods section). **b** A side projection; the vertical axis is aligned with that of gravity, as well as the channel depth, 0 μm marking the channel bottom plane; as the width of this panel spans a spatial range which is nearly six times longer than its height, for the sake of visualisation the vertical axis is stretched by 3/2; the *colour code* in the plot denotes time, which spans 4 s in this case. All pairs of tracers which were detected at some instant at a prescribed separation distance, $R_O$ = 10 ± 0.5 μm in this particular example, are collected to form one ensemble. The event at which the pair separation was nearest to $R_O$, marked by the *red circles* in the plot, is recorded as $t_O$ for the specific pair for later analysis. Each $R_O$ bin is 1 μm wide and centred at 6 through 50 μm, with sample sizes ranging from nearly $10^4$ to over $10^6$ pairs, respectively; sample size data are presented in Supplementary Fig. [6]. **c** A sub-sample of pair separation distances $R(\delta t)$ for 49 pairs belonging to the $R_O$ = 10 μm ensemble, presented on a semi-logarithmic scale; for each pair, $\delta t = t - t_O$ is the time elapsed since $t_O$. The *colour code* denotes time, scaled separately for each curve

flow is chaotic and 3D; second, the scales at which the dynamics takes place require the use of a microscope, where 3D imaging is non-trivial; third, the flow fluctuations in time dictate a high temporal resolution; and finally, the statistical nature of the problem demands a large sample of trajectories, which in turn requires long acquisition times and reliable automation.

To overcome these, we have implemented a novel method, which has been tested and presented in ref. [23]. In a nutshell, the 3D positions of the fluorescent particles are determined from a single camera two-dimensional (2D) imaging, by measuring the diffraction rings generated by the out-of-focus particle; this way the particle localisation problem turns into a ring detection problem, which is addressed accurately and efficiently in ref. [23]. By means of this direct Lagrangian particle tracking technique, we have established an experimental database[24] of about $10^7$ trajectories derived from passive tracers in elastic turbulence, generated in a curvilinear microfluidic tube; for further details see the Methods section.

In this letter, we report the results of pair dispersion due to the chaotic flow. Our data reveals that the memory of the initial relative velocity prevails the average dynamics, leading to a quadratic growth in time of the relative pair separation—the so-called 'Ballistic dispersion'—and shows no signature of the asymptotic exponential growth. In addition, we found that the relative velocity deviates from linear dependence on the separation distance already at about 8% of the tube width, indicating that the linear velocity assumption is violated for the most part of the motion, in contrast to the conceptual framework broadly used for the study of elastic turbulence.

## Results

**Establishing a statistically stationary elastic turbulence**. Let us consider a pair of passive tracers separated by the vector **R**; one realisation of such a pair is shown in Fig. 1a, b. The construction of the ensembles for the analysis to follow is outlined in Fig. 1, as well as in the Methods section.

As our intuition builds upon the common day-to-day experience with high Reynolds ($Re > 1$) flows, which are typically mixing, the chaotic nature of the trajectories presented in Fig. 1 may escape many readers. However, at the absence of polymers, the flow at low Reynolds ($Re < 1$) is laminar and regular, and tracers maintain their distance from the channel boundaries, exhibiting no crossing of trajectories; (Figs. 21–22 of ref. [9]) present the striking contrast between the laminar case of the pure solvent and the mixing elastic turbulence in the presence of polymers, both at low Reynolds.

Spatial features of the mean flow in our system, elastic turbulence in curvilinear microfluidic, can be revealed by transforming to the Eulerian frame of reference, as presented in Supplementary Fig. 2, and highlighted in its caption. These are in accordance with 2D Eulerian studies of statistically stationary fully developed elastic turbulence[5,25]. Despite some differences in the details of the experiments, this accordance should come as no surprise since the numbers characterising our flow, a Reynolds number smaller than $10^{-4}$ and a global Weissenberg number larger than 250 (see the Methods section), indeed, indicate that the results presented here were obtained in a regime lying well beyond the critical values for its statistical scaling properties to be Weissenberg and Reynolds dependent[5,25]; that is, in our experiment the Reynolds number is small enough to exclude any nonlinear effects due to inertia, and the Weissenberg number is large enough to achieve the elastic turbulence flow state which is both random in time and statistically time-independent.

A comparison of the local fluctuation intensities over time, as measured by the standard deviation fields, to the magnitude of mean velocity components, supports the notion of temporal randomness of our flow: this is most evident in the case of the non-stream-wise velocity components which fluctuate over time to a degree which exceeds that of the mean value in several regions across the pipe cross-sections, and comparable even to the stream-wise velocity component; Supplementary Fig. 2, specifically compare the values in sub-figures e to c and f to d. Realisations of velocity fluctuations in time, highlighting the randomness of the velocity field even at lower values of the Weissenberg number ($Wi$), have been shown in previous reports; see ref. [5], (Fig. 2) and Figs. 16–17 of ref. [25] (when comparing, note that our flow parameters should lead to a similar $Wi$ to the one in ref. [5], and are close to the $Wi = 679$ in ref. [25]; see the Methods section for clarification).

**Pursuing the asymptotic exponential pair dispersion**. Above we have recalled that random linear flows have been shown theoretically to result in an asymptotic exponential pair dispersion $\langle R^2(\delta t) \rangle = R_0^2 \exp[2\xi\,\delta t]$ (Supplementary Equation 3 in Supplementary Note 1), where the exponential rate $\xi$ is independent of the initial separation $R_0$; Supplementary Note 1 and references therein[11,18]. It is worth noting that $2\xi$, which can be identified with the second order generalised Lyapunov exponent, is not trivially related to the ordinary maximal Lyapunov exponent in the generic case; see (ref. [26], §3.2.1), (ref. [27], §5.3)[18], and others. The evaluation of the asymptotic exponential rate $\xi$ has drawn much attention in the literature; references to theoretical and numerical surveys can be found towards the end of Supplementary Note 1, while Supplementary Note 3 reviews the literature which follows from previous experimental studies. Our experimental data for $\langle R^2(\delta t) \rangle_{R_0} / R_0^2$ is presented in Fig. 2 on a semi-logarithmic scale; here, and in all that follows, $\langle \dots \rangle_{R_0}$ denotes ensemble averages differing by their initial

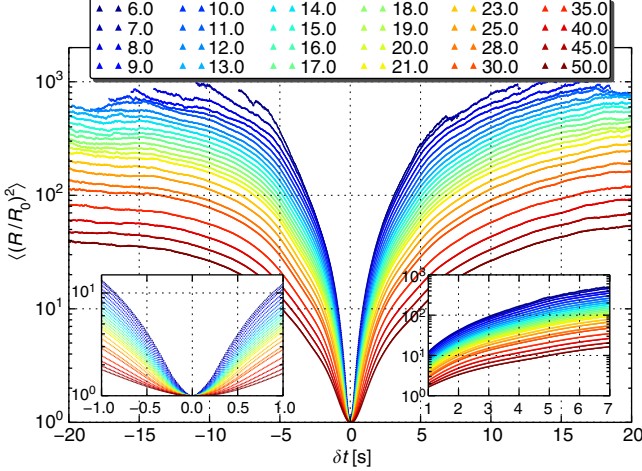

**Fig. 2** Pair dispersion normalised by the initial separation. The *plot* shows the average squared pair separation distance, normalised by the initial separation, $\langle (R(\delta t)/R_0)^2 \rangle_{R_0}$ for various $R_0$ between 6 and 50 μm; curves satisfying the asymptotic exponential pair dispersion $\langle R^2(\delta t) \rangle = R_0^2 \exp[2\xi\,\delta t]$, Supplementary Equation 3, would show-up on this semi-logarithmic presentation as straight lines, all sharing the same slope and, when extrapolated, hitting the origin, i.e., they should all collapse on a single linear relation. The *insets* present a zoom-in on the initial and intermediate temporal sub-intervals where the full range plot may seem to contain linear segments. Nevertheless, there is no unique slope which can be identified. Moreover, an exponential pair dispersion should extrapolate to the origin on this plot, which is clearly not the case here, and the curves do not merge asymptotically. The data show no supporting evidence for the exponential time dependence which follows Supplementary Equation 3. The un-normalised data $\langle R^2(\delta t) \rangle_{R_0}$ can be found in Supplementary Fig. 5

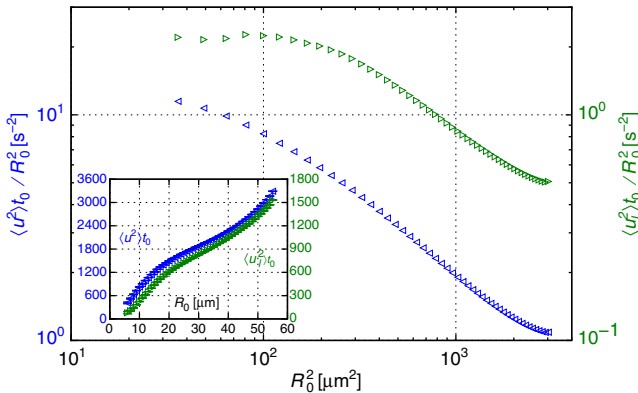

**Fig. 3** Initial relative velocity dependence on the separation distance. The second moments of the relative velocity $\langle u^2 \rangle_{R_0,t_0}$, (blue left-triangles) and the separation velocity $\langle u_l^2 \rangle_{R_0,t_0}$, (green right-triangle), where $u_l = \mathbf{u} \cdot \mathbf{R}/R$, are plotted in the inset (right axis values are half the left ones) as function of the initial separation distance $R_0$; both ensemble averages are taken at the initial time $t_0$, when the pairs separation distance is closest to $R_0$. Rescaling these data by the squared initial separation $R_0^2$ reveals the deviation from the commonly applied assumption of linear velocity field, as presented on a logarithmic scale in the main plot (right axis values are one order of magnitude smaller than the left ones). Had $\langle u^2 \rangle_R \propto R^2$ held, the rescaled curves would have remained constant; this is clearly not the case. Indeed, the $\langle u_l^2 \rangle_{R_0,t_0}/R_0^2$, data level off as $R_0$ approaches the smaller distances, providing supporting evidence for the linearity of $u_l$ with $R$ at scales smaller than 12 μm. However, this does not hold beyond a tenth of the channel depth. A linear flow regime is not supported by the rescaled relative velocity data $\langle u^2 \rangle_{R_0,t_0} R_0^2$, which values keep increasing even for the smallest $R_0$ values explored here. Further note that $\langle u^2 \rangle_{R_0,t_0}$ and $\langle u_l^2 \rangle_{R_0,t_0}$ (inset) are empirical estimators for the second order structure functions of the velocity and the longitudinal velocity, correspondingly; the former is the coefficient of the quadratic term in Eq. (2). The error bars in the inset (smaller than the marker) indicate the margin of error based on a 95% confidence interval

separations $R_0$. As discussed in the figure caption, our data show no supporting evidence for the exponential growth of $\langle R^2(\delta t) \rangle$.

**Failure of the linear flow assumption.** This raises questions regarding the extent to which elastic turbulence can be regarded as globally smooth, particularly in the presence of boundaries and mean flow. A velocity field consistent with linear flow behaviour would exhibit $\langle u_l^2 \rangle_R \propto R^2$ for the second order structure function of the longitudinal velocity $\langle u_l^2 \rangle_{R_0,t_0}$, where $\mathbf{u}$ denotes the relative velocity and $u_l = \mathbf{u} \cdot \mathbf{R}/R$; e.g., numerical simulations (Figs. 1 and 6 of ref. [28]). In our flow, clear deviations from linearity are evident already at separations beyond 12 μm, less than 10% of the width and depth of the microfluidic channel, as can be learnt from Fig. 3; a comparison to previous experimental results is drawn in Supplementary Note 2. The inset of Fig. 3 presents the mean squared relative velocities without rescaling; we shall return to these profiles soon.

**Relative pair dispersion.** Having not observed the exponential pair dispersion of long time asymptotics, and noting that the pairs of tracers we study explore also regimes where the linear flow assumption does not hold, we were still left with the puzzle of the nature of the qualitative similarity among the curves in Fig. 2 and its origin. Using a different data-derived quantity we have found that, for a significant fraction of the observation time, the mean relative pair dispersion evolves quadratically in time to leading order $\langle \|\mathbf{R}(\delta t) - \mathbf{R}_0\|^2 \rangle \propto \delta t^2$; this observation is evident in the insets of Fig. 4. To better understand the source for this scaling let

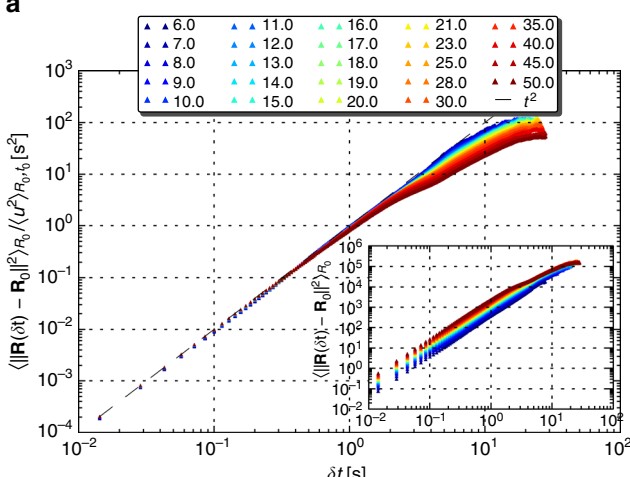

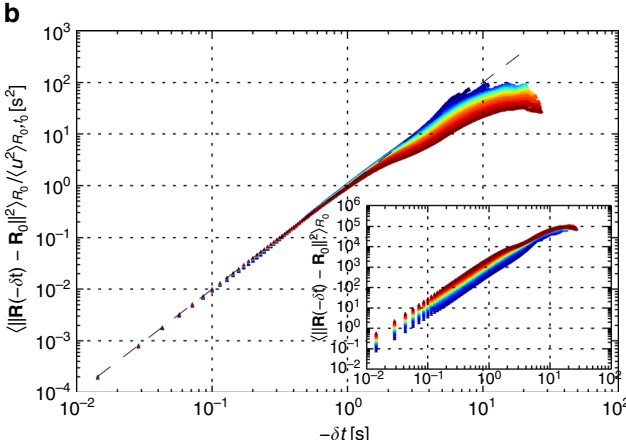

**Fig. 4** Relative pair dispersion forward and backwards-in-time evolutions. **a** Forward-in-time $\langle \|\mathbf{R}(\delta t) - \mathbf{R}_0\|^2 \rangle_{R_0}$ for various initial separations (inset) between 6 and 50 μm, collapse initially on a single curve which follows a power-law $\delta t^2$, once rescaled by the average squared relative velocity at the initial time, $\langle u^2 \rangle_{R_0,t_0}$. A significant deviation from $\delta t^2$ is noticed after 2–3 s, indicating the time beyond which higher order terms should be considered. **b** Backwards-in-time relative pair dispersion $\langle \|\mathbf{R}(-\delta t) - \mathbf{R}_0\|^2 \rangle_{R_0}$ for the same initial separations (inset), showing the same initial scaling collapse as the forward in time

us write the Taylor expansion around $\delta t = 0$

$$\mathbf{R}(\delta t) = \mathbf{R}_0 + \mathbf{u}_0 \delta t + \frac{1}{2} \dot{\mathbf{u}}_0 \delta t^2 + \mathcal{O}(\delta t^3). \quad (1)$$

Substituting this in the expression for the relative pair dispersion and considering the ensemble average over pairs of the same initial separation

$$\langle \|\mathbf{R}(\delta t) - \mathbf{R}_0\|^2 \rangle_{R_0} = \langle u^2 \rangle_{R_0,t_0} \delta t^2 + \langle \dot{\mathbf{u}} \cdot \mathbf{u} \rangle_{R_0,t_0} \delta t^3 + \mathcal{O}(\delta t^4), \quad (2)$$

we find that the leading order term at short times is indeed quadratic in $\delta t$—the so-called 'ballistic' regime.

**Establishing the case for the short-time statistics.** To test this further, we rescale the relative pair dispersion by the pre-factor, the mean initial squared relative velocity $\langle u^2 \rangle_{R_0,t_0}$. Unlike the case of inertial turbulence, for elastic turbulence there are no exact results nor scaling arguments to derive the coefficients appearing in Eq. (2). Therefore we extract them from the experimental data;

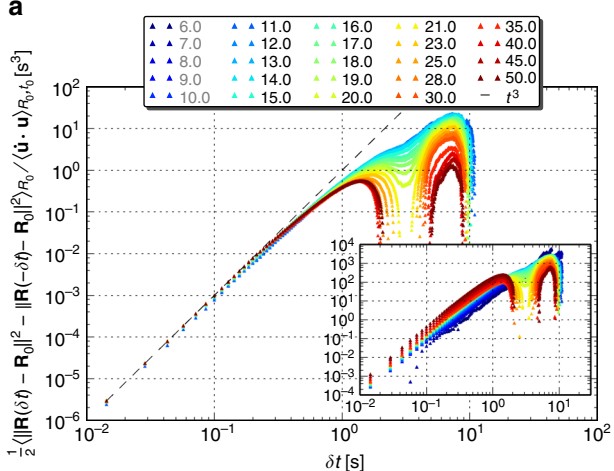

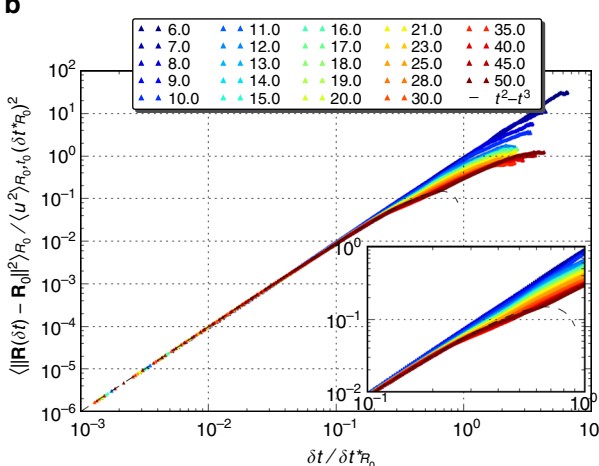

**Fig. 5** Relative pair dispersion time asymmetric terms and dimensionless form. **a** Taking the difference between the data sets plotted in the insets of Fig. 4, $\frac{1}{2}\langle \|\mathbf{R}(\delta t) - \mathbf{R}_0\|^2 - \|\mathbf{R}(-\delta t) - \mathbf{R}_0\|^2 \rangle_{R_0}$, exposes the contribution of the time asymmetric terms, odd powers in $\delta t$, presented here in the inset (sign inverted). Rescaling by the empirical estimator for $\langle \dot{\mathbf{u}} \cdot \mathbf{u} \rangle_{R_0,t_0}$, these data collapse on $\delta t^3$ initially; the data sets of $R_0 \leq 10$ μm (grey in the legend) are omitted from the main figure due to the scatter of the estimator; Supplementary Fig. 3. The plot shows a deviation from $\delta t^3$ at times shorter than 300 ms, indicating the dominance of higher order (odd) terms at early times and that the $\delta t^3$ term alone does not trivially explain the deviation from $\delta t^2$, observed in Fig. 4 after more than 2 s. **b** Rescaling the relative pair dispersion data (inset of Fig. 4a) by $\langle u^2 \rangle_{R_0,t_0} \left( \delta t^*_{R_0} \right)^2$ (see Eq. (2)), results in a dimensionless form, plotted here against dimensionless time, $\delta t$ rescaled by $\delta t^*_{R_0} = |\langle u^2 \rangle_{R_0,t_0} / \langle \dot{\mathbf{u}} \cdot \mathbf{u} \rangle_{R_0,t_0}|$; the empirical estimators of $\delta t^*_{R_0}$ can be found in Supplementary Fig. 4. The data sets indeed collapse onto a single curve $(\delta t/\delta t^*_{R_0})^2 - (\delta t/\delta t^*_{R_0})^3$, (dashed black line) for $\delta t/\delta t^*_{R_0} \lesssim 0.2$. The zoom-in (inset) emphasises the behaviour as $\delta t/\delta t^*_{R_0}$, approaches unity and the first two terms cancel out each other

see inset of Fig. 3. Indeed, we find that our data admits a scaling collapse with no fitting parameters, providing a convincing experimental evidence that these observations are well-described by the short-time expansion of the relative pair dispersion, exhibiting a significant deviation from $\delta t^2$ only after 2–3 s (Fig. 4).

Before discussing this timescale, we would like to first expose the sub-leading contributions to the initial relative pair dispersion. To this end, we subtract the backwards-in-time dynamics from the forward one. This way the time-symmetric terms, even powers of $\delta t$, are eliminated. The result, the time asymmetric

contributions presented in Fig. 5a, shows that indeed initially the next-to-leading order correction follows $\delta t^3$ and that the curves do collapse onto a single one when rescaled by $\langle \dot{\mathbf{u}} \cdot \mathbf{u} \rangle$, the appropriate coefficient in Eq. (2). The values of $\langle \dot{\mathbf{u}} \cdot \mathbf{u} \rangle_{R_0,t_0}$ were, once again, extracted from the experimental data (Supplementary Fig. 3).

However, the deviations from this scaling are noticeable earlier than half a second, much earlier than those from the ballistic behaviour discussed above. This hints that the later deviation observed in Fig. 4 is in fact due to higher order terms, potentially an indication of a transition to another regime. The observation that this transition takes place at an earlier time for the larger initial separations indicates the potential effects of the vessel size and its geometry. It may also be attributed to the limited range of the linear flow approximation, consistent with the data presented in Fig. 3.

**Exploring how far the short-time statistics apply.** Finally, let us consider the limitations of the relative pair dispersion short-time statistics description and its temporal range of application. The ratio of the first two coefficients in Eq. (2) constitute a timescale, $\delta t^*_{R_0} = \left| \langle u^2 \rangle_{R_0,t_0} / \langle \dot{\mathbf{u}} \cdot \mathbf{u} \rangle_{R_0,t_0} \right|$, which puts an upper bound for the ballistic approximation to be relevant. Rescaling Eq. (2) by $\langle u^2 \rangle_{R_0,t_0} \left( \delta t^*_{R_0} \right)^2$, the equation attains a dimensionless form, and one finds that the first two terms cancel each other as $\delta t/\delta t^*$ approaches unity due to the negative sign of $\langle \dot{\mathbf{u}} \cdot \mathbf{u} \rangle_{R_0,t_0}$, giving place to higher order terms to prevail the dynamics. Moreover, at that point, the expansion about the initial time is expected to fail altogether. The corresponding rescaled empirical data are presented in Fig. 5b; the empirical $R_0$ profile of $\delta t^*_{R_0}$ is provided in Supplementary Fig. 4.

## Discussion

On the one hand, our observations are consistent with the timescale $\delta t^*$, as sub-ballistic deviations from the $\delta t^2$ scaling are noticeable about $\delta t \approx 0.1\delta t^*$, as expected; see the zoom-in provided as the inset of Fig. 5, particularly for $R_0 \gtrsim 23$ μm, and to be compared with numerical simulations of inertial turbulence (ref. [29], Fig. 1). On the other hand, the data indicate that the relative pair dispersion remains near the $\delta t^2$ scaling even when $\delta t \approx \delta t^*$, which is remarkable and puzzling.

A question that may naturally come to mind is whether one could match the two limits, the short-time statistics and the long-time exponential prediction. Before making any further observations, one has to recall that the two are fundamentally different as the former is achieved by expanding about the initial time while the latter is attained as time approaches infinity, so attempting to match the two does not apply. Moreover, to demonstrate an exponential pair dispersion of the form of Supplementary Equation 3, it is necessary to show that the pair separation distance normalised by the initial separation, $\langle (R(\delta t)/R_0)^2 \rangle_{R_0}$, follows $\exp[2\xi\delta t]$ which is $R_0$ independent. Going back to Fig. 2, had our data supported the asymptotic exponential dispersion, the curves should have appeared as straight lines in this presentation, all having the same slope and, when extrapolated, hit the origin, collapsing all on a single linear relation. Our results clearly rule out the exponential dispersion regime in this case.

Before closing we must note that the short-time statistics, leading to ballistic dispersion, is a universal property which does not require any assumptions on the character of the flow. Thus far experimental[30] and numerical[29,31] results have been limited to the inertial subrange of high Reynolds number turbulence.

Beneath the dissipative scale a sign of this behaviour has been observed in simulations of inertial turbulence (ref. [31], Fig. 5).

And yet, to our knowledge the ballistic dispersion regime has not been discussed experimentally in the context of small scales chaotic flows, nor has it been confronted with the exponential pair dispersion prediction Supplementary Equation 3. On the contrary, reading recent publications on the subject, namely refs. [11,32,33], one may come to believe that the exponential dispersion has already been observed experimentally, while a closer examination reveals that this is not the case; further elaboration can be found in the Supplementary Note 3 and the conclusions therein.

We have demonstrated the predictive power of the ballistic dispersion in microfluidics elastic turbulence, a model system for a broader class of bounded chaotic flows at small scales.

## Methods

**Methods summary**. The work presented here relies on constructing a database of trajectories in an elastic turbulence flow[24]. Elastic turbulence is essentially a low Reynolds number and a high Weissenberg number phenomenon. The former means the inertial nonlinearity of the flow is over-damped by the viscous dissipation. The latter estimates how dominant is the nonlinear coupling of the elastic stresses to the spatial gradients of the velocity field compared with the dissipation of these stresses via relaxation. This is the leading consideration in the design of the flow cell.

The Lagrangian trajectories are inferred from passive tracers seeded in the fluid. In order to study the dynamics of pairs, the 3D positions of the tracers are needed to be resolved, even when tracers get nearby to each other. The requirement of large sample statistics dictates the long duration of the experiment, which lasts over days. The fluctuations due to the chaotic nature of the flow set the temporal resolution at milliseconds. This leads to a data generation rate of about 180 GB h$^{-1}$. Hence both the acquisition and the analysis processes are required to be steady and fully automated. The 3D positions of the fluorescent particles are determined using 2D single camera imaging, by measuring the diffraction rings generated by the out-of-focus particle. This way the particle localisation problem turns into a ring detection problem. To this end a new algorithm has been developed and tested[23]; the source is freely available online (https://github.com/eldad-a/ridge-directed-ring-detector).

**Microfluidic apparatus**. The experiments were conducted in a microfluidic device, implemented in polydimethylsiloxane elastomer by soft lithography, consisting of a curvilinear tube having a rectangular cross-section. The depth is measured to be 135 μm, the width is ~185 μm (Supplementary Fig. 1). The geometry consists of a concatenation of 33 co-centric pairs of half circles.

The working fluid consists of polyacrylamide (MW = $1.8 \times 10^7$ Da at mass fraction of 80 parts per million) in aqueous sugar syrup (1:2 sucrose to d-sorbitol ratio; mass fraction of 78%), seeded with fluorescent particles (1 micron Fluoresbrite YG carboxylate particles, PolySciences Inc.) at number density of about 50 tracers in the observation volume.

The flow is gravity driven.

**Physical considerations for the flow and passive tracers**. The viscosity of the Newtonian solvent, without the polymers, is estimated to be 1100 times larger than water viscosity at 22 °C. This leads to a polymer longest relaxation time of $\tau_p \simeq 100$ s[34], which is the longest timescale characterising the relaxation of elastic stresses in the solution. The ratio of the fluorescent particles mass density to that the working solution is about 0.75; Yet, the high viscosity of the working fluid and the small radius of the particles qualify them as passive tracers—the effects of buoyancy and inertia are essentially negligible as the terminal velocity is of the order of a tenth of a nanometre per second, and the inertia relaxation time is shorter than the tenth of a nanosecond. Additionally, for all practical purposes, we are allowed to neglect altogether contributions from Brownian motion to the dynamics of the fluorescent particles on the time scales over which they are observed—their diffusion coefficient leads to a variance increase of about a micron-squared in an hour.

Local velocity averaged over time in the Eulerian frame of reference showed a maximum over space of about $\max_{\mathbf{x}} \overline{v(\mathbf{x},t)} \simeq 250$ μm s$^{-1}$, for $v(\mathbf{x},t)$ denoting instantaneous local fluid velocity, here inferred from single particle trajectories, and time-averaging denoted by the bar. This results in a Reynolds number $Re \lesssim 10^{-4}$ and a global Weissenberg number $Wi = \tau_p \frac{\max_{\mathbf{x}} \overline{v(\mathbf{x},t)}}{\text{width}/2} \gtrsim 250$. To interpret these values in the light of ref. [25], one has to first match the manner by which $Wi$ is estimated. Plugging in the values provided in that report, using the maximal stream-wise velocity in (Fig. 10 of ref. [25]), in the definition we use above, one finds that the maximal $Wi$ used in ref. [25] would correspond to 447 in our case; using (Fig. 4 of ref. [25]), we can infer that the onset of developed elastic turbulence

corresponds to $Wi \simeq 165$, placing the parameters of our experiment in the regime of statistically stationary fully developed elastic turbulence.

**Imaging system**. The imaging system consists of an inverted fluorescence microscope (IMT-2, Olympus), mounted with a Plan-Apochromat 20× /0.8NA objective (Carl Zeiss) and a fluorescence filter cube; a Royal-Blue LED (Luxeonstar) served for the fluorophore excitation. A CCD (GX1920, Allied Vision Technologies) was mounted via zoom and 0.1× c-mount adaptors (Vario-Orthomate 543513 and 543431, Leitz), sampling at 70 Hz, 968 × 728 px, covering 810 × 610 μm laterally and the full depth of the tube. The camera control was based on a modification of the Motmot Python camera interface package[35], expanded with a home-made plug-in, to allow real-time image analysis in the RAM[23], recording only the time-lapse positions of the tracers to the hard drive.

**Lagrangian particle tracking**. To construct trajectories, the particle localisation procedure, introduced in ref. [23], has to be complemented by a linking algorithm. Here, we implemented a kinematic model, in which future positions are inferred from the already linked past positions. We used the code accompanying ref. [36] as a starting point. The algorithm was rewritten in Python (primarily using SciPy http://www.scipy.org/[37]), generalised to n-dimensions, the kinematic model modified to account for accelerations as well, a memory feature was added to account for the occasional loss of tracers, and it was optimised for better performance. The procedure accounts for the physical process of particles advected by a smooth chaotic flow and for the uncertainties. These arise from the chaotic in time nature of the flow ('physical noise') as well as from localisation and past linking errors ('experimental noise'). Finally, natural smoothing cubic splines are applied to smooth-out the experimental noise and estimate the velocities and accelerations[38,39]. The smoothing parameter is chosen automatically, where Vapnik's measure takes the role of the usual generalised cross-validation, adapted from the Octave splines package[40]. Links to the corresponding open-source Python code are provided below, under Data availability.

**Pairs analysis**. Within the trajectories database, we have identified pairs of tracers which were found at some instant at a separation distance close to a prescribed initial separation $R_0 = 6, 7, \ldots, 50$ μm, to within $\delta R_0 = \pm 0.5$ μm. The initial time $t_0$ for a trajectory was set by the instant at which the separation distance was closest to $R_0$. This way, each pair separation trajectory $R(\delta t)$ can contribute to an $R_0$ pairs ensemble at most once. See Fig. 1. The number of pairs considered in each $R_0$ ensemble is plotted in Supplementary Fig. 6 as function of $\delta t$.

Examining the ensemble averages of the relative separation velocity at the initial time $\langle u_l \rangle_{R_0,t_0}$, we do not find an indication that our sampling method introduces a bias for converging or diverging trajectories, at least for $R_0 \lesssim 22$ μm.

Our data support the linear flow approximation assumption at small enough scales, as indicated by the ensemble averages of the initial relative separation velocity; see Fig. 3 where $\langle u_l^2 \rangle_{R_0,t_0}/R_0^2$, (green right-triangles) levels-off at $R_0 \lesssim 12$ μm. The same regime is not reached for the relative velocity, yet the $\langle u^2 \rangle_{R_0,t_0}/R_0^2$ data in Fig. 3 (blue left-triangles) does not rule out this possibility for smaller scales.

**Data availability**. The data sets generated and analysed during the current study are available in the figshare repository, 10.6084/m9.figshare.5112991 [24].

All programming and computer aided analysis in this work relies on open-source projects; all based on tools from the SciPy ecosystem[41], primarily using IPython[42] as an interactive computational environment, Pandas[43] for data structures, and Matplotlib[44] for plotting.

Much of the source code developed in the course of this study is available as open-source at: https://github.com/eldad-a/ridge-directed-ring-detector; https://github.com/eldad-a/particle-tracking; https://github.com/eldad-a/natural-cubic-smoothing-splines

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

## Acknowledgements

We thank A. Frishman for the helpful and extensive discussions of the theory and O. Hirschberg for useful discussions of the mathematical and statistical analysis; E.A. had fruitful discussions with J. Bec, S. Musacchio, D. Vincenzi, EW Saw and R. Chetrite, kindly organised by the latter; both authors gained from thorough discussions with V. Lebedev, as well as the helpful reading and comments of an earlier version of the manuscript by G. Boffetta, A. Celani and M. Feldman. This work is supported by the Lower Saxony Ministry of Science and Culture Cooperation (Germany; grant #VWZN2729) and the Israel Science Foundation (ISF; grant #882/15).

## Author contributions

V.S. proposed the study of pair dispersion in elastic turbulence. E.A. designed the experiment, performed the measurements and analysed the results. Both authors discussed the results, the relevant literature, and wrote the manuscript.

## Additional information

**Competing interests:** The author declares no competing financial interests.

