## [Peer Review File · Nature Communications]

Reviewers' comments:

Reviewer #1 (Remarks to the Author):

The authors have studied how pairs of particle separate in a model “turbulent” flow, which in this case is associated with the low Reynolds number flow of an elastic fluid. The methods indicate that one Reynolds number is selected ($Re \ll 1$) and that one Weissenberg number is selected. The authors indicate a new type of imaging method for collecting a large amount of statistics about the flow – I am not able to judge the method so I will assume it works and has been checked and tested. The principal result claimed by the authors is that the pair dispersion is governed by ballistic features of the flow. I suppose it fair to say that I am not expert enough to judge the basis of the claim. Below I give a number of questions that seemed important to me as I tried to assess the claims and data presented in the paper. I even spent some time trying to read some of the other papers from this group in the literature (e.g. the citations in the references) but it was not always clear to me – as the authors themselves bring up in the paper – that the results/conclusions are always consistent with previous publications/claims from the group. Some more discussion and clarification will help. I believe this paper may be reporting a significant new result but I think a good revision is necessary to better make the case the authors are trying to present.

Questions:

1) For example, won't a Taylor series expansion of the exponential argument lead to equation (2) over some time scale? How important is it that the approximate relations (e.g. equation 2) work over some time interval and if so what limits the time interval? Surely some features of the (fluctuating?) flow should be relevant to establishing the necessary time interval. I did not find any discussion of this point though it may be my ignorance of the field.

2) Figure 1 is presented as typical of the pair trajectories. Admittedly, these independent realizations did not look chaotic to me – in fact they looked very regular as if one was sampling different streamlines in a 3D laminar flow. I do not recall the authors reporting any “control” experiments, e.g. a laminar viscous flow in this rather highly curved geometry with a viscous fluid to achieve a similarly low Reynolds number with a Newtonian fluid. Without such a control I think it is difficult to appreciate how much of what is being imaged is a consequence of the suggested “elastic turbulence” and how much might be do to 3D flow features induced by the highly curved geometry (I understand that the flow given be complicated as such elastic turbulence has been described) – I think several necessary “controls” need to be reported to properly set the flow and dynamics in perspective.

3) Finally, are there some estimates for the typical time scale of velocity fluctuation in “elastic turbulence”? Should this time scale be relevant to the kinds of measurements reported in the figures.

Additional remarks:

a) On p. 2, in the expression for the exponential pair separation, what is $\bar{\zeta}$ in the argument of the exponential function? Point the reader to the SI where there is some discussion.

b) On p. 2: “Recall that random linear flows can be shown to result in an asymptotic exponential pair dispersion” – a reference is appropriate. For example, I do not know this result.

c) Figure 1: I do not understand why “The figure demonstrates the chaotic nature of the flow as manifested by pairs”. All we see are two trajectories separating. But in any given realization, particles can be on different streamlines so the separations in time could be a little different. This relates to question (2) above as to my (possibly naïve) eye figure 1 is too regular to be the result of different realizations of a chaotic flow, though this assessment likely needs more informative on the time scale

of the fluctuations.

d) Figure caption 1b: "the vertical axis is stretched by 3/2" – so the vertical axis on the image needs clarification since the axes do not indicate any difference in scaling. In any event, this is very unclear as presented.

e) p. 6: "also regimes were where the linear flow assumption does not hold"

f) p. 6: "let us consider the applicability limitations of the relative" -> re-word

g) p. 9, top: "corresponding" -> corresponding

h) p. 9: "And yet, to our knowledge it has not been discussed experimentally in the context of small scales chaotic flows" -> to what does "it" refer?

Reviewer #2 (Remarks to the Author):

The authors have accumulated an impressive data set for the motions of tracer particles in a dilute polymer suspension, which is driven at max speed ~250um/s through microfluidic channels. They find a ballistic scaling of the pair dispersion, in apparent contraction with prevailing theory.

Although the study appears to have been carefully performed and the experimental details are described in detail, I have several concerns that prevent me from recommending the present manuscript version for publication in Nature Communications.

The authors may be able to address these concerns by performing additional experiments and data analysis so I would like encourage them to submit an improved version.

Major comments/questions:

1. If Fig. 1a,b show representative tracer trajectories, then it is not evident that the suspension has reached a statistically stationary state of elastic turbulence.

The bends at the ends arise from the projection and the trajectories are otherwise quite similar compared to those expected of Brownian tracer particles in standard Poiseuille flow. Can the authors offer additional experimental evidence confirming that their system is in a truly chaotic static?

2. An important difference compared with previous studies of elastic turbulence appears to be the presence of a strong externally imposed (gravity driven) flow field, which could delay the onset of elastic turbulent effects. The data analysis and the results provided in the manuscript appear to suggest that the system may be in a Poiseuille-like flow state with a weak superimposed turbulent flow component. This would directly explain the observed ballistic scaling. The authors' data should allow them to reconstruct and present

- (i) the mean velocity profiles across the channel cross-section and
- (ii) the corresponding velocity fluctuation maps.

Such additions data would help convince the reader that the system is indeed in a chaotic flow state.

3. I propose that authors perform and present data from a control experiment with a simple fluid (without the polymers, but same flow rates/channels/tracers), to demonstrate

a significant statistical difference compared to the elastic turbulence regime.

4. What is the direction of gravity in Fig 1a? What are the relative densities of tracer particles and fluid?

Minor corrections/comments:

Supplementary figure 4: do the authors mean "between 6 and 50 μm " ?

The authors should consider including the content of Suppl. Notes 1 & 2 in the main text, as these are relevant for understanding the state of the field.

REVIEWERS' COMMENTS:

Reviewer #1 (Remarks to the Author):

The authors describe a detailed quantitative study of the separation of pairs of particles in elastic turbulence (a low Reynolds number chaotic flow of a polymeric solution at high Weissenberg number). The revised paper has introduced various edits that make it easier to read and understand for the nonspecialist. As this reviewer is on shaky grounds in his understanding of the statistical characterization of turbulent flows, I am taking the authors on faith in several of their statements, which is fine. The paper is very well written. The figures are very good and the text, in general, describes well the figures and their interpretation. It is a pleasure to read such a paper and think about the results, which seem original, unexpected and very well characterized. The main point of the paper is that using a recently developed imaging method for tracking particles and performing statistics on particle pairs the authors compare their results for the pair dispersion with the exponential separation expected for random (chaotic) flows. The data do not have this feature so the authors instead question the "spatial smoothness" of the velocity field, and they write that their results show that the spatial smoothness assumption breaks at scales smaller than a tenth of the system size. I do wonder whether perhaps the flow is not truly "random" and that perhaps the dynamics of these complex polymeric instabilities might be the origin of the discrepancy – of course this might be equivalent to a breakdown of "spatial smoothness" but the authors do not appear to address this other way to address a failure in the traditional hypotheses for exponential separation. Perhaps the authors can add some remark to their paper. Otherwise, I think this paper will make an excellent contribution to the literature and will likely generate a conversation among researchers in the field.

Additional remarks:

1) The opening sentence of the introduction is rather poorly chosen in my view: "To truly appreciate how come even fluid dynamics experts find elastic turbulence astonishing, we first have to realise that our intuition is based on scenarios where the flow is dominated by inertia, quantified by high values of the Reynolds number." – Surely there are fluid dynamics experts who are aware of low Reynolds number flows and not simply only knowledgeable about flows at high Reynolds numbers. GI Taylor published a movie in the 1960s on these flows that is very well known and there are many books on the topics. The authors should find a different, less misleading way to open their paper.

2) p. 1: "So long as the driving force and the boundary conditions are steady, so will be the flow. It is for this reason that mixing in microfluidic devices is normally limited to diffusion." – Read naively the sentence sounds true but it is well known in the field to be false. Steady three-dimensional laminar flows can mix well in some cases. This was demonstrated by Stroock et al. in a paper in Science already in 2002 (reference 12 of the present paper) but was known in theory to the fluid dynamics community already in the late 1980s due to the work of Ottino and colleagues (and was likely known in some circles before that). I suggest some re-wording.

3) p. 3: "Weissenberg and Reynolds dependent" -> do the authors mean "Weissenberg number dependent and Reynolds number independent"? If the details were dependent on both parameters, than simply changing the Reynolds numbers has an effect and it would be important to know what value of (low) Reynolds number triggers the effect? This confuses me since I associate "elastic turbulence" with a low Reynolds number (no critical value) but some critical Weissenberg number. Perhaps I am misreading what the authors have written or perhaps they can revisit their description.

4) The data shows no supporting evidence for the exponential growth of the pair dispersion and from

this the authors ask questions regarding the extent to which elastic turbulence can be regarded as globally smooth; this leads to their main claim. I want to ask one more question about the logic, as indicated in my opening paragraph. The existing theory apparently is that random linear flows result in an asymptotic exponential pair dispersion. So if the experimental measurements disagree with the theory, rather than claiming a breakdown in spatial smoothness of the velocity field, is it not also possible that the flow field is not completely random in time? Might not these unstable polymeric flows have time dependence perhaps tied to the shear rate and the relaxation times of the polymeric solutions? Of course this might be equivalent to a breakdown of "spatial smoothness" but the authors do not appear to address this other way to address a failure in the traditional hypotheses for exponential separation.

Reviewer #2 (Remarks to the Author):

The authors have done a very good job in carefully revising their paper, and addressing and answering the concerns/questions raised in my previous report. The additional supplementary material has helped improve the manuscript further and readers can now better evaluate the differences between normal Poiseuille flow and the viscoelastic flow. I think this revised version is suitable for publication and I only have two additional minor comments.

1) It may help broaden the scope of the paper if the authors could add a brief discussion explaining how their results compare to/differ from ordinary Taylor dispersion, see e.g. *Science*, 354(6317), 1252-1256 (2016). Is it possible to estimate by how much Taylor dispersion can be enhanced through the addition of polymers? This could be very interesting from an application perspective.

2) Several figures have unlabeled color bars. I recommend to add both physical units and descriptive labels to all of these.

In reply to Reviewer #1 of
Manuscript NCOMMS-16-15717

We are delighted to learn that Reviewer #1 appreciates the importance of our new results and insights which followed.

A major consideration in choosing Nature Communication as our venue is to reach out to an audience much broader than our own scientific community, as we believe the lessons we have learnt will prove valuable to many. Hence, despite the fact that Reviewer #1 does not consider himself “*expert enough to judge*”, his critique is indispensable for us. It highlights what was lacking in our presentation, prior knowledge we mistakenly took for granted.

Therefore, we are certain that by fully addressing his comments and suggestions, as we believe we have, the quality of the manuscript has substantially improved. Consequently, we are confident it is now ready for publication in Nature Communication.

The major questions raised by Reviewer #1, as we understand his thorough report, can be categorised under two topics:

- (i) Time scales associated with the asymptotic expressions and relating them.
- (ii) Establishing the case that our study is indeed of statistically stationary developed elastic turbulence, strongly mixing, and distinct from laminar and regular flows.

To address the first point:

There is no timescale over which a Taylor expansion of the exponential dispersion expression can lead to the Ballistic dispersion (Eq. 2); the revised discussion now includes a new paragraph which shows this. Moreover, as a specific example, if one assumes an exponential dispersion taking place per realisation and plugs it as an ansatz to Eq. 1, then into Eq. 2, one is then led to identify the exponential rate (ξ) with the mean squared initial velocity ($\langle u^2 \rangle$), resulting in two contradictions to the exponential pair dispersion prediction when confronted with our data: (a) the exponential rate turns out to be dependent on the initial separation R_0 ; and (b) the exponential rate must be negative, in the light of Fig. 5 for example (as well as Fig S3). Once again, this is a specific example and we decided not to include it in the paper as the revised manuscript already includes a general mathematical argument as well as empirical ones, which are derived from our data.

As to the importance of the limited time interval for the Ballistic dispersion to hold, its origin and reasons for limitation: the mathematical explanation of the Ballistic dispersion as a Taylor expansion at short times provided us with the time scales, as presented and discussed in the manuscript. Thus on the one hand this shows a quantitative agreement with the data, supporting our main claim for

short-time statistics (rather than long time asymptotics). Additionally, the limited time scale indeed appears to affect the first odd-order term as expected (see Fig. 5 in the manuscript). On the other hand, the literature focuses on exponential pair dispersion in the form of Eq. S3 which, to our knowledge has yet to be observed; this should not be surprising once one realises how restrictive the underlying assumptions are, limiting the generality its in practice, as admitted by Batchelor¹ himself; see quotes toward the end of SI Note 3.

Reviewer #1 also asked:

“3) Finally, are there some estimates for the typical time scale of velocity fluctuation in “elastic turbulence”? Should this time scale be relevant to the kinds of measurements reported in the figures.”

The longest relaxation time, denoted in the Methods section by τ_p , is estimated at about 100s, which a property of the polymeric solution; estimates are based on Liu et al (2009)². The Eulerian correlation time of velocities was reported to be comparable to the longest relaxation time, based on Jun & Steinberg (2011) where a similar setup was used. The time scales discussed in our paper are much shorter and depend on the initial separation R_0 (see Fig. S4), as expected from the mathematical analysis we present in the paper from which $\delta t_{R_0}^*$ is derived. The time scales associated with fluctuations, which can be derived from the root-mean-squared spatial gradient of velocities in Jun & Steinberg (2011) are subseconds, which are much shorter when compared with the ones in our paper. Therefore we concluded that these are not trivially related to ours. The point raised by Reviewer #1 is an interesting one, and may be related to the Lagrangian correlation time, which was not accessible prior to creation of our dataset. However, this is far beyond of the scope of the current presentation and is left for future research.

As for the second point:

Thanks to the report by Reviewer #1 we have realised that in writing we regretfully did not build well enough the reader’s intuition as to what to expect from a laminar flow, and to eventually truly appreciate the striking features of elastic turbulence.

In the revised manuscript we have now carefully addressed this, providing explicit arguments and new SI Figures, as well as referring to the relevant literature: Figs. 21-22 in Groisman & Steinberg (2004)³ (appended below) contrast the mixing properties of the dilute polymer solution to that of the pure solvent (containing no polymers) both in a curvilinear channel, as well as to Jun & Steinberg (2011)⁴, specifically Fig. 2 therein, showing the profile of the streamwise component of the velocity across the

¹ Batchelor, G. K. The effect of homogeneous turbulence on material lines and surfaces. *Proceedings of the Royal Society A: Mathematical, Physical and Engineering Sciences* 213, 349–366 (1952)

² Liu, Y., Jun, Y. & Steinberg, V. Concentration dependence of the longest relaxation times of dilute and semi-dilute polymer solutions. *Journal of Rheology* 53, 1069–1085 (2009)

³ Groisman, A. & Steinberg, V. Elastic turbulence in curvilinear flows of polymer solutions. *New Journal of Physics* 6, 29 (2004).

⁴ Jun, Y. & Steinberg, V. Elastic turbulence in a curvilinear channel flow. *Phys. Rev. E* 84 (2011).

curvilinear channel, to be contrasted with the new Fig S2 (g) and (h), showing the same features as Fig. 10 in Jun & Steinberg (2011).

For the sake of completeness, please see attached a figure comparing side-by-side trajectories taken in our system with and without polymers.

In what follows we provide additional point-by-point response to the comments raised.

We are grateful for the meticulous report by Reviewer #1, which helped us in better delivering the lessons we would like to share with the wide scientific and engineering community.

Several additional revisions and point-by-point response:

- ✓ Background on Exponential dispersion, theory and experiments
 - ✓ References added to the introduction, earlier in the text to address: *On p. 2: “Recall that random linear flows can be shown to result in an asymptotic exponential pair dispersion” – a reference is appropriate. For example, I do not know this result.*
 - ✓ Where *the exponential rate* (ξ) first appears, the main text has a cross-reference to SI Note 1, where the theoretical background is provided. We have also addressed the relation to the generalised Lyapunov exponents in the main text itself. Additionally, ξ and $\tilde{\xi}$ have been exchanged hoping to achieve better clarity (tilde denoting the fluctuating quantity while the former is reserved for the asymptotic result).
- ✓ Establishing that our study is of Steady State well-developed elastic turbulence:
 - ✓ With regards to the requested control experiments using a viscous Newtonian fluid:
 - ✓ The introduction (lines #35-37) now refers the reader to Figs. 21-22 in Groisman & Steinberg (2004) , also referred to where Fig.1 is cross-referenced in the revised paper, as well as in its caption; for convenience, the Figures are attached in what follows.
 - ✓ Where appropriate (e.g. lines #72-87), the revised text refers the reader to Jun & Steinberg (2011) where the requested control experiments have been reported. As a matter of fact, our case is far more conservative as the Reynolds in that report was three orders of magnitude larger.
 - ✓ [Rev #1 report:] “*The methods indicate that one Reynolds number is selected ($Re \ll 1$) and that one Weissenberg number is selected*” ; The justification for this choice is provided in a new paragraph (lines #79-87).
- ✓ Further comments and Typos [(d)-(h) in the report by Reviewer #1]
 - ✓ All addressed in the text as highlighted by Reviewer #1

Figures referred to in the revised manuscript:

Groisman, A. & Steinberg, V. Elastic turbulence in curvilinear flows of polymer solutions. *New Journal of Physics* 6, 29 (2004).

Figure 21. Schematic drawing of the curvilinear channel showing the inlet, a region of observation and the outlet.

Figure 22. Photographs of the flow taken with the laser sheet visualization (figure 21) at different N . The field of view is 3.07 mm \times 2.06 mm, and corresponds to the region shown in figure 21 (rotated 90° counterclockwise). Bright regions correspond to high concentration of the fluorescent dye. (a) Flow of the pure solvent at $N=29$; (b–d) flow of the polymer solution at $Wi=6.7$ and at $N=8, 29, 54$, respectively.

Jun, Y. & Steinberg, V. Elastic turbulence in a curvilinear channel flow. *Phys. Rev. E* 84 (2011)

FIG. 2. Longitudinal velocity profiles across a channel of a solvent laminar flow in a curvilinear channel at various pressure drops (starting from small values at bottom to top; $r=0$ corresponds to the inner channel wall).

FIG. 10. The entire mean longitudinal velocity profile across the channel of a polymer solution flow for $Wi=951$.

Polymer solution vs pure-solvent: elastic instabilities compared with laminar flow.

Figure reproduced from E. Afik's Feinberg Graduate School PhD progress report; Data taken before the development of the ridge directed ring detector algorithm [Afik (2015)⁵], before severe limitations on detection quality were lifted (as manifested by the broken polygon-like tracks). The figure compares trajectories of tracers in the presence (right) and absence (left) of polymers. Colours represent velocity magnitude, rescaled per trajectory (blue is slowest, red is fastest). This is a 30 seconds arbitrary data extract.

Fastest laminar trajectories among the ones presented move at about 160 $\mu\text{m/s}$;

Fastest polymeric trajectories among the ones presented move at about 100 $\mu\text{m/s}$

Both solutions consist of 80% sugar and 80ppm polyacrylamide in the Non-Newtonian solution. The channel is 120 μm and 240 μm internal and external radii; hence the estimated Weissenberg is $Wi \approx 166$, which is just at the transition from the elastic instability regime to the onset of developed elastic turbulence; see the Methods section (new sub-section, titled "Physical considerations: fluid, flow and tracers") in the revised manuscript for how this is estimated.

Nevertheless, it evident that the Newtonian fluid is laminar, where tracers maintain their distance from the boundaries, a feature broken by the elastic instability. The results reported in the manuscript were taken at a flow well-beyond the onset of elastic turbulence, based on the algorithm that was developed for that purpose, to overcome the shortcomings of the algorithm based on which the data in the above figure was taken, namely it allowed us to increase the Wi and the number density of the tracers, and yet achieves a much higher detection rate, even when tracers are nearby, as required for the study of pair dispersion.

⁵ Afik, E. Robust and highly performant ring detection algorithm for 3d particle tracking using 2d microscope imaging. *Scientific Reports* 5, 13584 (2015).

In reply to Reviewer #2 of
Manuscript NCOMMS-16-15717

We were very content to learn that Reviewer #2 appreciates the complexity and quality of our experimental work as well as the analysis which followed. The report by Reviewer #2 highlights several points where we evidently did not provide the reader with the information needed to fully understand the work and the lessons which it teaches.

We are therefore grateful for the helpful report by Reviewer #2. We would also like to thank him for encouraging us to submit a revised version of the manuscript, bringing it to a higher level prior to its publication in Nature Communications.

The report by Reviewer #2 as we understand it raises two major concerns. The first invites us to demonstrate that we are indeed in a statistically stationary state of well-developed elastic turbulence. The second indicates that we should better present the departure from Poiseuille-like flow. Additionally, in his report, Reviewer #2 proposes the use of control experiments to highlight the differences between the flow of the dilute polymer solution we used and the pure solvent (in the absence of polymers), compared under the same conditions.

All the above have been fully addressed in the revised manuscript. We now provide further details which demonstrate the system is indeed in the regime of well-developed elastic turbulence (for the specific modifications, please see the point-by-point reply below); the new SI figures, which follow from the analysis proposed by Reviewer #2, support this and show good accordance with the relevant literature, that is, Jun & Steinberg (2011)¹; specifically Fig. 2 in that reference, showing the profile of the streamwise component of the velocity across the curvilinear channel, to be contrasted with the new Fig S2 (g) and (h), exhibiting the same features as Fig. 10 in Jun & Steinberg (2011) (the above mentioned Figs. 2 & 10 are attached below). In addition, the above together with the rest of the new Figure S2 exhibits a clear departure from Poiseuille-like (or any laminar) flow.

We have also provided new references, Figs. 21-22 in Groisman & Steinberg (2004)² (appended below), contrasting the mixing properties of the dilute polymer solution to that of the pure solvent (containing no polymers) both in a curvilinear channel, as well as to those in Jun & Steinberg (2011); it is worth noting that the studies reported in Groisman & Steinberg (2004) and Jun & Steinberg (2011), as well as others, all included mean flows, as characteristic of any turbulent flow in an open system (whenever there is a net mass flux through the system).

While all the above consist the controls for our experiments, for the sake of completeness, please see attached a figure comparing side-by-side trajectories taken in our system with and without polymers.

Finally, we carefully revised the manuscript following further comments provided in the report, introducing further information, as requested. Following Reviewer #2 recommendation, we have included content from the first two SI Notes in the main text where appropriate. To help the reader focus on the main findings and conclusions, we have decided to maintain the full versions of these sections as SI Notes.

¹ Jun, Y. & Steinberg, V. Elastic turbulence in a curvilinear channel flow. *Phys. Rev. E* 84 (2011).

² Groisman, A. & Steinberg, V. Elastic turbulence in curvilinear flows of polymer solutions. *New Journal of Physics* 6, 29 (2004).

In what follows we have collected point-by-point responses to the comments raised in the report by Reviewer #2, which we consider of very high value as it helped us realise what needed improvement to better deliver the lessons we would like to share with the wide scientific and engineering community.

Several Additional revisions and point-by-point response:

- ✓ Establishing the case for statistically stationary well-developed elastic turbulence:
 - ✓ With regards to the requested control experiments using a viscous Newtonian fluid:
 - ✓ The introduction (lines #35-37) now refers the reader to Figs. 21-22 in Groisman & Steinberg (2004), also referred to where Fig.1 is cross-referenced in the revised paper, as well as in its caption; for convenience, the Figures are attached below.
 - ✓ Where appropriate (e.g. lines #72-87), the revised text refers the reader to Jun & Steinberg (2011) where the proposed control experiments have been reported. As a matter of fact, our case is far more conservative as the Reynolds in that report was three orders of magnitude larger.
 - ✓ It is worth noting that the diffusion coefficient of our tracers is below half a nanometer-squared per sec (due to the high viscosity). This is so small that the contribution of Brownian diffusion to the translation of the tracers we used in our liquid, after even an hour, is estimated at less than a micron (estimated by the Stokes-Einstein relation); the observation time per tracer in our experiment is typically less than a minute.
 - ✓ The second of the above mentioned two new paragraphs (lines #79-87) also provides the statements which show that our study was conducted in a flow which is statistically stationary developed elastic turbulence; we have incorporated further explicit arguments to the Methods section (new sub-section, titled “Physical considerations: fluid, flow and tracers”).
 - ✓ Reviewer #2 suggested further analysis into the mean flow and its fluctuations; this has led to the introduction of the new Fig S2, which is referred to in the main text and a comparison is drawn to laminar and regular flows, supported by Jun & Steinberg (2011), Figs. 2 and 10 therein (attached below).
- ✓ Further comments and Typos [in the report by Reviewer #2]
 - ✓ The direction of gravity is clarified in the caption of Fig. 1(b), where the vertical axis of panel (b) is aligned with gravity; another phrase has been added the caption of Fig. 1(a), stating that gravity points out of that panel.
 - ✓ Under “Physical considerations: fluid, flow and tracers” in the Methods section the relative density of the tracers is provided, along with arguments which show why these tracers qualify as passive tracers (as buoyancy and inertia are negligible; above we explained why Brownian diffusion is irrelevant).
 - ✓ Fig S4 typo addressed in the text, as highlighted by Reviewer #2

Figures referred to in the revised manuscript:

Groisman, A. & Steinberg, V. Elastic turbulence in curvilinear flows of polymer solutions. *New Journal of Physics* 6, 29 (2004).

Figure 21. Schematic drawing of the curvilinear channel showing the inlet, a region of observation and the outlet.

Figure 22. Photographs of the flow taken with the laser sheet visualization (figure 21) at different N . The field of view is $3.07 \text{ mm} \times 2.06 \text{ mm}$, and corresponds to the region shown in figure 21 (rotated 90° counterclockwise). Bright regions correspond to high concentration of the fluorescent dye. (a) Flow of the pure solvent at $N=29$; (b–d) flow of the polymer solution at $Wi=6.7$ and at $N=8, 29, 54$, respectively.

FIG. 2. Longitudinal velocity profiles across a channel of a solvent laminar flow in a curvilinear channel at various pressure drops (starting from small values at bottom to top; $r=0$ corresponds to the inner channel wall).

FIG. 10. The entire mean longitudinal velocity profile across the channel of a polymer solution flow for $Wi=951$.

Polymer solution vs pure-solvent: elastic instabilities compared with laminar flow.

Figure reproduced from E. Afik's Feinberg Graduate School PhD progress report; Data taken before the development of the ridge directed ring detector algorithm [Afik (2015)³], before severe limitations on detection quality were lifted (as manifested by the broken polygon-like tracks). The figure compares trajectories of tracers in the presence (right) and absence (left) of polymers. Colours represent velocity magnitude, rescaled per trajectory (blue is slowest, red is fastest). This is a 30 seconds arbitrary data extract.

Fastest laminar trajectories among the ones presented move at about 160 $\mu\text{m/s}$;

Fastest polymeric trajectories among the ones presented move at about 100 $\mu\text{m/s}$

Both solutions consist of 80% sugar and 80ppm polyacrylamide in the Non-Newtonian solution. The channel is 120 μm and 240 μm internal and external radii; hence the estimated Weissenberg is $Wi \approx 166$, which is just at the transition from the elastic instability regime to the onset of developed elastic turbulence; see the Methods section (new sub-section, titled "Physical considerations: fluid, flow and tracers") in the revised manuscript for how this is estimated.

Nevertheless, it evident that the Newtonian fluid is laminar, where tracers maintain their distance from the boundaries, a feature broken by the elastic instability. The results reported in the manuscript were taken at a flow well-beyond the onset of elastic turbulence, based on the algorithm that was developed for that purpose, to overcome the shortcomings of the algorithm based on which the data in the above figure was taken, namely it allowed us to increase the Wi and the number density of the tracers, and yet achieves a much higher detection rate, even when tracers are nearby, as required for the study of pair dispersion.

³ Afik, E. Robust and highly performant ring detection algorithm for 3d particle tracking using 2d microscope imaging. *Scientific Reports* 5, 13584 (2015).

In reply to Reviewer #1 of Manuscript NCOMMS-16-15717A

Reviewer #1 (Remarks to the Author):

The authors describe a detailed quantitative study of the separation of pairs of particles in elastic turbulence (a low Reynolds number chaotic flow of a polymeric solution at high Weissenberg number). The revised paper has introduced various edits that make it easier to read and understand for the nonspecialist. As this reviewer is on shaky grounds in his understanding of the statistical characterization of turbulent flows, I am taking the authors on faith in several of their statements, which is fine. The paper is very well written. The figures are very good and the text, in general, describes well the figures and their interpretation. It is a pleasure to read such a paper and think about the results, which seem original, unexpected and very well characterized. The main point of the paper is that using a recently developed imaging method for tracking particles and performing statistics on particle pairs the authors compare their results for the pair dispersion with the exponential separation expected for random (chaotic) flows. The data do not have this feature so the authors instead question the "spatial smoothness" of the velocity field, and they write that their results show that the spatial smoothness assumption breaks at scales smaller than a tenth of the system size. I do wonder whether perhaps the flow is not truly "random" and that perhaps the dynamics of these complex polymeric instabilities might be the origin of the discrepancy - of course this might be equivalent to a breakdown of "spatial smoothness" but the authors do not appear to address this other way to address a failure in the traditional hypotheses for exponential separation. Perhaps the authors can add some remark to their paper. Otherwise, I think this paper will make an excellent contribution to the literature and will likely generate a conversation among researchers in the field.

Additional remarks:

1) The opening sentence of the introduction is rather poorly chosen in my view: "To truly appreciate how come even fluid dynamics experts find elastic turbulence astonishing, we first have to realise that our intuition is based on scenarios where the flow is dominated by inertia, quantified by high values of the Reynolds number." - Surely there are fluid dynamics experts who are aware of low Reynolds number flows and not simply only knowledgeable about flows at high Reynolds numbers. GI Taylor published a movie in the 1960s on these flows that is very well known and there are many books on the topics. The authors should find a different, less misleading way to open their paper.

2) p. 1: "So long as the driving force and the boundary conditions are steady, so will be the flow. It is for this reason that mixing in microfluidic devices is normally limited to diffusion." - Read naively the sentence sounds true but it is well known in the field to be false. Steady three-dimensional laminar flows can mix well in some cases. This was demonstrated by Stroock et al. in a paper in Science already in 2002 (reference 12 of the present paper) but was known in theory to the fluid dynamics community already in the late 1980s due to the work of Ottino and colleagues (and was likely known in some circles before that). I suggest some re-wording.

3) p. 3: "Weissenberg and Reynolds dependent" -> do the authors mean "Weissenberg number dependent and Reynolds number independent"? If the details were dependent on both parameters, than simply changing the Reynolds numbers has an effect and it would be important to know what value of (low) Reynolds number triggers the effect? This confuses me since I associate "elastic turbulence" with a low Reynolds number (no critical value) but some critical Weissenberg number. Perhaps I am

misreading what the authors have written or perhaps they can revisit their description.

4) The data shows no supporting evidence for the exponential growth of the pair dispersion and from this the authors ask questions regarding the extent to which elastic turbulence can be regarded as globally smooth; this leads to their main claim. I want to ask one more question about the logic, as indicated in my opening paragraph. The existing theory apparently is that random linear flows result in an asymptotic exponential pair dispersion. So if the experimental measurements disagree with the theory, rather than claiming a breakdown in spatial smoothness of the velocity field, is it not also possible that the flow field is not completely random in time? Might not these unstable polymeric flows have time dependence perhaps tied to the shear rate and the relaxation times of the polymeric solutions? Of course this might be equivalent to a breakdown of "spatial smoothness" but the authors do not appear to address this other way to address a failure in the traditional hypotheses for exponential separation.

Point-by-Point Response

We are grateful for the meticulous reports by Reviewer #1, which helped us in better delivering the lessons we would like to share.

- 1) In his first remark, the Reviewer highlights an alternative interpretation of our introduction, one which had not occurred to us. The introduction opening sentence and the closing sentence of the 3rd introduction paragraph have been revised accordingly and are now less ambiguous.
- 2) The revised 2nd paragraph now includes 'chaotic mixers', which renders it more precise, thanks to the Reviewer's remark. We have also added "chaotic in time" to the 3rd paragraph to contrast our flow from the temporally steady flows of these 'chaotic mixers'.
- 3) Reading the Reviewer's report leads us to believe that he understands correctly the Re-Wi parameter space region for elastic turbulence; to be on the safe side, we have added a clarification at the end of the relevant paragraph.

4) This point makes us realise we have skipped some necessary background; therefore, once again, we are grateful for the Reviewer's indispensable comments. To establish the case of a random in time flow, the revised manuscript includes a new paragraph, the 3rd under the subheading "Establishing a statistically stationary elastic turbulence" in the Results Section; this paragraph makes the following arguments:

- a) Our data shows intense temporal fluctuations. This is evident, for example, in the Eulerian picture when examining the ratio of standard deviation to mean value of velocities presented in the Supplementary Fig 2, subfigs e&f compared to c&d respectively.
- b) Temporal randomness of the velocity field has been shown to prevail in similar geometries at even lower Weissenberg numbers than ours; in this respect the reader is referred in the main text to the Eulerian velocity power spectrum, Fig. 2 in Groisman & Steinberg (2001)¹, as well as in representative time series published in Figs. 16–17 of Jun & Steinberg (2011)².

Finally, had the smoothness assumption held for our flow, a temporally steady flow would have been simple laminar shear flow, This is ruled out by the Eulerian picture presented in the Supplementary Fig 2.

¹ Groisman, A. & Steinberg, V. Efficient mixing at low reynolds numbers using polymer additives. Nature 410, 905–908 (2001).

² Jun, Y. & Steinberg, V. Elastic turbulence in a curvilinear channel flow. Phys. Rev. E 84 (2011)

In reply to Reviewer #2 of Manuscript NCOMMS-16-15717A

Reviewer #2 (Remarks to the Author):

The authors have done a very good job in carefully revising their paper, and addressing and answering the concerns/questions raised in my previous report. The additional supplementary material has helped improve the manuscript further and readers can now better evaluate the differences between normal Poiseuille flow and the viscoelastic flow. I think this revised version is suitable for publication and I only have two additional minor comments.

1) It may help broaden the scope of the paper if the authors could add a brief discussion explaining how their results compare to/differ from ordinary Taylor dispersion, see e.g. *Science*, 354(6317), 1252-1256 (2016). Is it possible to estimate by how much Taylor dispersion can be enhanced through the addition of polymers? This could be very interesting from an application perspective.

2) Several figures have unlabeled color bars. I recommend to add both physical units and descriptive labels to all of these.

Point-by-Point Response

We are delighted that our revised manuscript has received such a positive review, and we are thankful to the Reviewer for his previous and current constructive reports.

- 1) The work by Aminian *et al.* (2016)³, suggested by the Reviewer, analyses and discusses effective diffusion in a laminar steady channel flow; the authors extend the original work of Taylor (1953)⁴ by considering how Taylor Dispersion is affected by the cross-sectional aspect ratio of the channel, and employing modern methods. While

³Aminian, M., *et al.* How boundaries shape chemical delivery in microfluidics. *Science*, 354(6317), 1252-1256 (2016).

⁴Taylor, G. Dispersion of soluble matter in solvent flowing slowly through a tube. *Proc. R. Soc. A*. 219, 1137 (1953).

Aminian *et al.* study the outcomes of diffusion combined with a laminar flow, our work studies the dispersion of tracers in a random flow where particle diffusion is negligible (see “Physical considerations for the flow and passive tracers” under the Methods section in the manuscript); Aminian *et al.* characterise their system using the Péclet number, which evaluates in their case to the order of 10^3 , while ours exceeds their highest by over 5 orders of magnitude; while the diffusive timescale characterising their system is less than a 15 minutes, ours goes beyond 6 months. Finally, we note that Aminian *et al.* do not discuss efficient mixing; as a matter of fact, their system of interest is an excellent example to what we had in mind when we concluded the second paragraph of the introduction by: “mixing in microfluidic devices is normally limited to diffusion”. In our study, efficient mixing consists an essential property of the system; it is in such scenarios that the lessons learnt in our work are of higher interest and value.

- 2) All colour bars now have unit dimensions. The observables denoted by colours are explained in the corresponding figure captions; including these as descriptive labels would have required either the introduction of a new notations (which in turn need further explanations in the figure caption) or quite a bit of text for clarity; this is why we have decided to present it in the current form.